# Label-Free Flow Cytometry: A Powerful Tool to Rapidly and Accurately Assess the Efficacy of Chemical Disinfectants

**DOI:** 10.3390/microorganisms13051156

**Published:** 2025-05-19

**Authors:** Andreea Pîndaru, Luminița Gabriela Măruțescu, Marcela Popa, Claude Lambert, Mariana-Carmen Chifiriuc

**Affiliations:** 1Department of Botany and Microbiology, Faculty of Biology, University of Bucharest, 030018 Bucharest, Romania; pindaru.andreea@s.bio.unibuc.ro (A.P.); carmen.chifiriuc@bio.unibuc.ro (M.-C.C.); 2Research Institute of University of Bucharest, University of Bucharest, 030018 Bucharest, Romania; marcela.popa@bio.unibuc.ro; 3Neurotoxicology, Development and Bioactivity, LCOMS/ENOSIS, Université de Lorraine, 57000 Metz, France; doctor.lambert@gmail.com

**Keywords:** label-free flow cytometry, disinfectant efficacy testing, viable non-culturable bacteria

## Abstract

A rapid and accurate evaluation of a chemical disinfectant’s bactericidal efficacy is crucial for ensuring effective infection control, preventing the spread of pathogens, and supporting the development of new disinfectant formulations. In this study, we report a rapid, label-free flow cytometry (FCM) protocol for evaluating the bactericidal efficacy of disinfectants. Five commercial disinfectants (alcohols, oxidizing agents, and alkylating agents) were evaluated against type strains recommended by EN 13727+A2 and ten clinical strains. The label-free FCM method allowed the determination of disinfectant efficacy through assessment of scatter light profiles (FSC-H/SSC-H) and count differences. The label-free FCM provided the results in approximately 4 h and showed strong correlation with standard tests (91.4%, sensitivity 0.94 and specificity 0.98) that can take up to 48 h. Our results represent a proof-of-principle that label-free FCM can reliably assess the efficacy of chemical disinfectants, the same day, and substantially faster than the current growth-based methods. Additionally, the study highlights the potential of the FCM method for detecting the occurrence of viable but non-culturable bacteria following treatment with chlorine-based disinfectants. With its speed, accuracy, and capability to identify bacterial injuries at a single-cell level, the FCM method is a powerful tool for assessing the efficacy of new disinfectant formulations.

## 1. Introduction

Traditional culture-based methods have long served as the cornerstone for detecting and quantifying bacterial contamination in clinical and environmental settings. However, these methods suffer from several key limitations: they are time-consuming, often requiring 24–72 h to yield results [1]; they fail to detect bacteria in a viable but nonculturable (VBNC) state [2,3]; and they may underestimate microbial load in samples subjected to chemical disinfection or environmental stress [4]. These limitations can lead to false-negative results, delayed interventions, and incomplete risk assessments. In contrast, flow cytometry (FCM) enables rapid, high-throughput analysis of microbial populations at the single-cell level, with the capability to distinguish live, dead, and damaged cells based on structural and functional markers [5,6,7]. Label-free FCM approaches further improve the single-cell analysis by removing the requirement for fluorescent staining, which simplifies sample preparation [8,9]. By leveraging these advantages, our study introduces and evaluates a label-free FCM protocol for the assessment of chemical disinfectant efficacy, offering a timely and more accurate alternative to conventional methods.

Chemical disinfectants are widely used to reduce bacterial bioburden on various surfaces in medical environments. However, clinically important pathogens have been found to persist even after disinfection, posing a risk of infection [10,11,12,13]. Several studies, both experimental and real-world, have demonstrated bacteria’s ability to adapt to chemical disinfectants [14,15]. Outbreak investigations reveal that nosocomial pathogens can withstand exposure to disinfectants such as peracetic acid [16], quaternary ammonium compounds [17], and glutaraldehyde [8]. For instance, *Serratia marcescens* was not eliminated by a quaternary ammonium compound-based disinfectant, and *Mycobacterium massiliense* strains responsible for outbreaks in 38 hospitals in Rio de Janeiro, Brazil, exhibited resistance to glutaraldehyde, which was used for endoscope disinfection. These findings underscore the need for the careful selection of disinfectants, routine environmental screening, and, when necessary, testing disinfectant tolerance in specific contexts [12,16,18,19,20,21].

The issue of bacterial resistance to chemical disinfectants remains a critical yet unresolved challenge in infection control. Current knowledge is largely derived from in vitro studies, which predominantly report increases in minimum inhibitory concentrations (MICs). Decreased susceptibility and acquired resistance have been documented across a range of disinfectants, including quaternary ammonium compounds, alcohols, chlorine, hydrogen peroxide, and iodophors (Table 1). Resistance mechanisms are often nonspecific, involving efflux pumps and alterations in membrane structure or function [21] (Table 1). While MIC values observed under laboratory conditions are often significantly lower than the in-use concentrations of these agents, raising questions about their clinical relevance, notable exceptions exist. For instance, the TriABC efflux pump has been shown to confer triclosan resistance in *Pseudomonas aeruginosa* [22]. These findings underscore the need for more clinically relevant studies to clarify the true impact of disinfectant resistance in healthcare settings.

The current approach for disinfectant efficacy evaluation relies on culture-based assays that test selected microorganisms in either their planktonic state (suspension testing) or on a surface (carrier testing) after a predetermined contact time. These methods, standardized under EN 13727+A2 [39], detect microbial growth on solid media. However, even when using fast-growing species like *E. coli*, results take approximately 48 h. Most importantly, studies have shown that chemical disinfectants can induce a viable but non-culturable (VBNC) state in which bacteria remain alive but fail to grow on routine culture media. This state presents a significant challenge to infection control and disinfection practices, as VBNC bacteria can evade detection by conventional methods while retaining the potential to resuscitate and regain pathogenicity under favorable conditions. Table 2 summarizes reports on viable but non-culturable pathogens identified following disinfection. In contrast to conventional methods, single-cell technologies such as flow cytometry (FCM) offer rapid analysis within minutes, require minimal sample volumes, and provide faster and more accurate detection of bacterial viability. Given the need for rapid evaluation of disinfectants during emergencies such as outbreaks, a rapid efficacy testing method that delivers reliable results comparable to standard suspension tests would be beneficial. In addition, FCM can provide important insights into the heterogeneous bacterial response to chemical disinfectants. Studies have demonstrated that FCM can detect bacteria in a VBNC state that display tolerance to chemical disinfectants [23]. For example, *Klebsiella pneumoniae* has been found to persist in a VBNC state on surfaces for at least a month [40].

Our prior study demonstrated that the label-free FCM protocol can be used to assess the efficacy of quaternary ammonium compounds [9]. In the present study, we describe a label-free FCM protocol to rapidly evaluate the bactericidal efficacy of various chemical disinfectants, with different mechanisms of action, which are commonly used in clinical settings, including alcohols, oxidizing agents, and alkylating agents. Furthermore, we explored the potential of FCM combined with fluorescent dyes for the detection of a disinfectant-induced VBNC state that standard efficacy tests cannot detect.

## 2. Materials and Methods

### 2.1. Chemical Disinfectants

All chemical disinfectant products were purchased from local distributors. Four products were ready to use, and one was in the form of a concentrate to be diluted for the working solution. The active agents are listed in Table 3. The disinfectants were tested at concentrations and contact times recommended by the manufacturers. Disinfectant dilutions were prepared using sterile distilled water (according to the European standard).

### 2.2. Bacterial Isolates, Growth and Culture Conditions

For disinfectant efficacy testing, a total of 14 bacterial strains were used, comprising 4 reference strains *Staphylococcus aureus* ATCC 6538, *Pseudomonas aeruginosa* ATCC 15442, *Enterococcus hirae* ATCC 10541 and *Escherichia coli* K12 NCTC 10538 and 10 clinical bacteria belonging to the ESKAPEE group [*K. pneumoniae* (*n* = 1), *E. coli* (*n* = 2), *Acinetobacter* (*n* = 2), *P. aeruginosa* (*n* = 2), *S. aureus* (*n* = 2); *E. faecium* (*n* = 1)]. The clinical strains included in this study were selected from a previously characterized Romanian strain collection (RADAR project (02/07/2018–30/06/2022)—Selection and dissemination of antibiotic resistance genes from wastewater treatment plants into the aquatic environment and clinical reservoirs, PN-III-P4-ID-PCCF-2016-0114). The clinical strains were recovered from different sources of infection (Table 4) and the majority (9/10) were multidrug resistant (MDR), defined as resistant to three or more antibiotic classes. Frozen glycerol stock cultures of the clinical and reference strains were streaked on Plate Count Agar (Scharlau 01-161-500) (PCA) and incubated for 18–24 h at 35 ± 2 °C. For each assay, fresh bacterial cultures (16–18 h) were used for preparation of suspensions with a density of ~10^8^ CFU/mL, in Tryptone buffer.

### 2.3. Standard Qualitative/Quantitative Suspension Tests

To assess the efficacy of the selected disinfectants, we used qualitative and quantitative suspension tests according to standard SR EN 13727+A2. Antiseptics and chemical disinfectants, Quantitative testing of the suspension for the evaluation of bactericidal efficacy in the medical field, test method and requirements (phase 2, step 1)] [39]. We have adapted it for testing in micro-volumes, using 96-well microtiter plates [54]. A volume of twenty microliters of planktonic bacteria (~10^7^ CFU/mL) was added to each of the disinfectant’s two-fold dilutions prepared in water and gently homogenized (~10^6^ CFU/mL). After the manufacturer’ specified contact time, a volume of twenty microliters of the mixtures were transferred to 180 µL of Neutralizing Fluid (Scharlau 02512500, Barcelona, Spain) supplemented with 5% Tween 80 (Scharlau TW0080100) for 5 min. An aliquot of twenty microliters was subsequently added to 180 µL of Tryptic Soy Broth (Scharlau 02-200-500) (TSB) and incubated for 18–24 h at 35 ± 2 °C. The minimum inhibitory concentrations (MICs), i.e., the lowest disinfectant concentrations inhibiting bacterial growth, were determined by spectrophotometry (620 nm) and by visual readings. A volume of ten microliters from each sample test and control was plated on PCA plates. The disinfectant concentrations corresponding to PCA plates with no bacterial growth were recorded as MBCs (minimum bactericidal concentrations) values.

Additionally, to assess quantitatively the efficacy of disinfectants, suspension tests were performed. Briefly, bacterial suspensions were exposed to serial two-fold disinfectant dilutions, at a contact time specified by the manufacturer. At the end of the neutralization step, the bacterial counts were determined using the FCM protocol and standard suspension tests. Logarithmic reductions in viable counts were determined as described by SR EN 13727+A2. A reduction in viable counts of ≥ 5-log was considered sufficient for bactericidal efficacy. The experiments were performed in triplicate. Each assay included a positive control with no treatment, and negative controls with no bacterial cells added.

### 2.4. Disinfectant Efficacy Testing Using Label-Free FCM

The principle of rapid disinfectant efficacy testing using label-free FCM is based on detecting changes in bacterial cell scatter properties and count rates (events/s) following disinfectant exposure. Further, these FCM measurements can be used to predict MBC values and determine the efficacy of chemical disinfectants. The experimental workflow is depicted in Figure 1.

#### 2.4.1. Sample Preparation for Label-Free FCM

The FCM protocol was used to evaluate the bactericidal effect of the chemical disinfectants using scattered light signal characteristics and count rates [9,55]. The bacteria were exposed to serial dilutions of chemical disinfectant solutions. At the end of the contact time, the disinfectant action was neutralized with Neutralizing Fluid (Scharlau 02512500) supplemented with 5% Tween 80 (Scharlau TW0080100) for 5 min. Afterwards, twenty microliters of the mixtures were transferred to 180 µL TSB and incubated for 3.5–4 h at 35 ± 2 °C. At the end of incubation time, the disinfectant-treated samples and untreated controls were washed with filtered saline solution (10,000× *g*, 5 min), suspended in 1000 μL filtered saline solution and transferred to a polystyrene round base flow cytometry tube. The experiments were performed in three independent repeats.

#### 2.4.2. Detection of Bacteria Using Label-Free FCM

Our previous work demonstrated that flow cytometry (FCM) can be reliably used to detect bacteria treated with antimicrobial agents after a 4 h incubation period [56]. To validate this hypothesis, we tested a collection of 42 clinical bacterial strains (see Appendix A) along with 4 reference strains. Fresh bacterial cultures grown on Plate Count Agar (Scharlau 01-161-500) were used to prepare standard suspensions in tryptic soy broth (TSB). Aliquots of 20 μL of each suspension were further aseptically transferred to 180 µL of TSB (~10^6^ CFU/mL). Bacterial growth was then measured after 4 h using FCM in three independent experiments. Samples were acquired using the FCM settings described below.

#### 2.4.3. Instrument Settings

All the tests were carried out using a BD AccuriC6 Plus™ flow cytometer, equipped with a 488 nm blue laser (detections FL1 533/30 nm, FL2 585/40 nm, FL3 > 670 nm) and a 640 nm red laser (FL4 675/25 nm). Single-color controls were used to locate bacterial populations and determine compensation settings. The BD^®^CS&T (cat 661414) tracking beads were used for calibration. The acquisition rate was set at 35 µL per minute and acquired for ≥25 µL.

#### 2.4.4. Gating Strategy and Data Analysis

Most bacteria range in size between 500 and 1000 nm and conventional FCM instruments are able to detect and count them as single particles [9,55,57]. The untreated samples were measured before disinfectant-treated samples, and the acquired FCM data were used to define the “Bacteria” gate for each strain. The fluorescent dye SYTO9, a cell-permeant marker, was used to identify the SYTO9+ population corresponding to the control samples, in a side scatter (SSC) vs. green fluorescence plot. This population was then back-gated onto a forward scatter (FSC) vs. SSC plot to establish the “Bacteria” gate. Then, unlabeled disinfectant-treated samples were measured and the scattered light events falling outside the “Bacteria” gate were excluded from quantitative analysis. Doublet discrimination was performed using a plot SSC-area versus SSC-H. A threshold of 1000 was set on FSC-H based on control data, ensuring the detection of all bacterial cell events while minimizing background noise. The events that had increased area, without linear relationship to the height, were considered as doublets and excluded. Subset gating was the final step in the gating process. The resulting bacterial count rates (events/s) corresponding to each disinfectant dilution were normalized by scaling them to their respective negative control (no bacteria) and positive control (growth control) resulting in modified bacterial counts as shown by Filbrun [55].

### 2.5. Assessment of Disinfectant-Induced VBNC State in Bacteria Using FCM

Various chemical disinfectants were shown to induce sub-lethal injury to certain bacteria, rendering them non-culturable [9,58,59]. Samples treated with disinfectant at MBC and ½ MBC were analyzed using FCM with fluorescent labelling to assess the potential disinfectant-induction of VBNC state in bacteria. The LIVE/DEAD^®^ Bac Light™ (Thermo Fisher Scientific, Waltham, MA, USA) kit was used according to the manufacturer’s recommendations. Unstained samples, inactivated bacteria (treated at ~90 °C for 30 min), and single stain controls were included for each analysis. Bacterial cells with altered membrane were stained in red with propidium iodide (PI) and considered to be dead cells, whereas bacterial cells with an intact membrane were stained in green with SYTO9TM. Staining procedures were all conducted in the dark and following the manufacturer’s recommendations. Each acquisition included an unstained sample to define the positive staining gate and single stain controls. For each sample, 1.5 μL of PI (20 nM in DMSO) and 1.5 μL of SYTO9^TM^ (3.34 mM in DMSO) were added, and then the samples were incubated for 15 min in the dark, before FCM analysis. The detectors FL1 (533/30) for SYTO9^TM^, and FL3 (670LP) for propidium iodide (PI) were used for fluorescence measurements. The FL1 and FL3 voltages were adjusted so that the unstained bacteria were set within the first order of the logarithmic scale of fluorescence. The stained controls and samples were then analyzed on these settings. For each sample, up to 5000 events were acquired. The cytometry data were analyzed using the AccuriC6™ Plus analysis software.

### 2.6. Statistical Analysis

MBC values are reported as median values. The results obtained with the label-free FCM protocol were compared to those of the standard suspension tests. Discrepancies of classification were calculated as percentages of error. *p*-values were calculated using unpaired *t*-test with Welch’s correction and a threshold of 0.05. The assumption of normality was assessed prior to applying the unpaired *t*-test with Welch’s correction. The receiver operating characteristic (ROC) analysis of counts/s falling in the “Untreated bacteria” gate was used to predict disinfectants’ MBC values.

## 3. Results

### 3.1. Minimum Inhibitory and Bactericidal Concentrations

All tested disinfectants proved to be effective against tested bacteria when using standard working concentrations and contact time specified by the manufacturer (Table 5). The disinfectants’ MIC values determined by qualitative suspension tests were similar to disinfectants’ MBC values from quantitative suspension tests. The median of the MBC value determinations is presented in Table 5. Notably, Clor2Klin showed the highest efficacy (lowest MBC values) against the planktonic bacteria. This trend was observed regardless of the bacterial species. The superior performance of Clor2Klin may be attributed to its stabilized chlorine dioxide formulation, which exerts a potent oxidative effect that disrupts bacterial cell membranes, proteins, and nucleic acids [60]. For all the disinfectants tested, the efficacy tests showed no significant differences between reference strains versus clinical isolates.

### 3.2. Detection of Bacteria Using the FCM Assay

After 4 h, FCM was able to detect, based on scatter light signals, a distinct cell population, readily separated from background noise. The counts for this population were significantly increased for each of 46 bacterial strains tested (Appendix A) (*p* < 0.0001) (Figure 2B). Therefore, the incubation time was set at 4 h for all subsequent experiments to record the increase in number of bacterial cells after exposure to different disinfectant concentrations. A typical scattered light pattern of bacterial growth is depicted in Figure 2A.

### 3.3. Disinfectant Efficacy Using the Label-Free FCM Method

A total of 14 bacterial strains (Table 4) were tested against the five chemical disinfectants (Table 3) with results available within 4 h. The principle of disinfectant efficacy using the label-free FCM protocol is that following disinfectant exposure, neutralization and culture step of 4 h at 35 ± 2 °C, the bacteria will exhibit both altered scatter profiles and count rates (events/s). These measurements can be converted into disinfectant MBC values. The untreated samples were measured before the disinfectant treated samples. The data acquired (scatter profiles and counts) for each bacterial strain tested were used to set the “untreated bacteria” gate (Figure 3). This gate was further inherited by all disinfectant-treated samples in the assay. The FCM counts were recorded for each disinfectant-treated sample. Further, the obtained FCM data were normalized by scaling them against their respective negative control (uninoculated samples) and control (untreated samples), resulting in adjusted bacterial counts, as demonstrated by Filbrun [55].

At or above bactericidal concentrations, flow cytometry (FCM) data revealed alterations in the multiparametric distributions of disinfectant-treated samples, displaying distinct susceptibility-associated patterns compared to the untreated population (bounded in red). We observed a strong correspondence between the disinfectant concentrations that induced these efficacy-associated signatures across 14 bacterial strains and their respective bactericidal concentrations (above MBC or at MBC values). Figure 4 illustrates representative results for Gram-negative (Figure 4A) and Gram-positive (Figure 4B) strains. The absence of such patterns can be interpreted as indicative of a lack of bactericidal activity.

Although the MBC cannot be directly inferred from this visualization method, the predicted MBC consistently falls within the range of tested disinfectant concentrations. This finding supports further quantitative analysis of FCM-derived cell counts to predict the MBC. To validate this approach, we compared FCM counts from disinfectant-treated samples with outcomes from standard suspension tests. Figure 4 illustrates the FCM count analysis as compared with standard growth-based method for Gram-negative (Figure 4A) and Gram-positive (Figure 4B) strains, as representative results. For each tested disinfectant, across all 14 strains, significant differences were detected in FCM counts between untreated controls, disinfectant samples treated at MBC, and those exposed to sub-inhibitory concentrations (Kruskal–Wallis, *p* < 0.001; Figure 5).

Further, the FCM counts were paired with the standard test results and converted into “above or MBC” or “not MBC”. An ROC curve was plotted to determine the cut off points of FCM counts in relation to disinfectant efficacy determined by standard suspension tests. As determined by the ROC curve, the FCM counts/s can be classified as “above or MBC” or as “not MBC” using the empirically set cut-off of <0.1 counts/s (Figure 6). The area under the ROC curve was 0.96, with a standard error 0.010, 95% confidence interval (0.94–0.98), *p* < 0.0001. When this threshold was applied to the tested samples, 64 out of 70 FCM results (91.4% of FCM assay results) were classified correctly for “above or MBC” status (sensitivity 0.94 and specificity 0.98) (Table 6 and S2, Figure 6).

The lowest disinfectant concentration in the series causing a decrease of <0.1 in counts/s were determined by FCM as the MBC value. Based on the threshold of <0.1 counts/s, a total of six samples were deemed as “above or MBC” by FCM and “not MBC” by standard suspension tests. Agreement between the novel FCM method and standard tests was 98.5% over 69 results, with MBC values falling for each method within ±1 doubling dilution.

At or above bactericidal concentrations, the FCM data (the FSC/SSC plots) revealed alterations in the scatter signals of disinfectant-treated samples, indicating susceptibility-associated patterns compared to the untreated population (bounded in red). The FCM counts determined within the untreated bacteria gate were used to predict the disinfectant MBC values. Quantitative analysis of FCM-derived cell counts (tables) as compared with standard suspension test results is presented (see corresponding Appendix A). The FCM counts/s were normalized for each sample in series and matched with standard suspension test results and categorized as “≥MBC” or “not MBC”. *E. faecium* (Ef16) and *E. coli* (Ec2) are presented as demonstrative examples. The fcs files are included in the Appendix A and are publicly available at https://community.cytobank.org/cytobank/experiments/115822 (accessed on 24 April 2025) (FCM for bactericidal efficacy of disinfectants). Cytobank community is a platform that allows researchers to annotate, analyze, and share results along with the underlying single-cell data [61].

### 3.4. Disinfectant-Induced VBNC Bacteria Determined by FCM with Fluorescent Labelling

Disinfectants, such as chlorine or ammonium compounds, have been implicated in inducing the VBNC state in microorganisms [3,62,63,64]. The disinfectant-treated samples, at MBC and ½ MBC, were analyzed using FCM with LIVE/DEAD™ BacLight bacterial viability kit staining in order to assess whether the tested disinfectants were able to induce the VBNC state in bacteria. Manual gating was performed to quantify the presence of live/dead cells (Figure 7) and % fluorescence. Viable bacteria were identified in disinfectant-treated samples at sub-inhibitory concentrations. The presence of double-stained cells (membrane-compromised cells) or VBNC populations were determined for *E. coli* treated with Clor2Klin^TM^, at ½ MBC. Figure 7 depicts the population of VBNC bacteria, as determined using the double staining.

## 4. Discussion

Chemical disinfectants are extensively used in health-care settings to prevent the transmission of pathogens in patients. These products must be used judiciously and their impact on target pathogens should be assessed and monitored closely in order to continue using them for the control of pathogen spread, especially in clinical settings. Moreover, some disinfectants may cause irritation and trigger inflammatory responses. While this study focuses on bactericidal activity, the assessment of immunological aspects needs to be considered for further evaluation of disinfectant safety. These assays could target the MAPK signaling pathway that is activated by disinfectants, particularly in the context of infection and tissue stress [65]. Additionally, the pro- and anti-inflammatory cytokines could represent useful biomarkers. IL-37, for example, is a key anti-inflammatory cytokine with the potential to limit excessive immune activation during infection [66].

This study advances the label-free FCM method to accelerate and to provide a more in-depth assessment of chemical disinfectants’ efficacy. Using a total of five commercial chemical disinfectants with different modes of action, we have demonstrated that the label-free FCM protocol readily differentiates between disinfectant’s MBC and ½ MBC, within the equivalent of 4 h. The principle of label-free flow cytometry (FCM) involves detecting changes in scatter properties and count rates of bacterial cells after disinfectant treatment. This method allows the determination of disinfectant efficacy through quantitative assessment of scatter light profiles (FSC-H/SSC-H) and counts the differences in bacterial cells exposed to varying concentrations of disinfectant. At bactericidal concentrations, the samples were found to exhibit a drastic alteration of both scatter light profiles and counts compared to untreated samples, indicating that the cell wall was damaged as a result of disinfectant treatment according to the modes of action of these antimicrobial agents. Chemical disinfectants act on the cell wall leading to disruption of the membrane and consequent leakage of intracellular contents and subsequent bacterial death [21].

Further, these FCM measurements can be used to predict the MBC values and determine the efficacy of chemical disinfectants. The FCM counts determined within the “Bacteria” gate were classified as “above or MBC” or “not MBC”, following pairing with the standard test results. Then, using the ROC curve, we determined that using a threshold of <0.1 counts/s, the label-free FCM method showed excellent prediction of efficacy of alcohols, oxidizing and alkylating agents. The label-free FCM method and standard test results agreed with over 91.42% (64/70) of all measurements (sensitivity 0.94 and specificity 0.98). Based on data of point-to-point analysis, the MBC values determined using the label-free FCM method were slightly higher than those determined using the standard assay. The FCM method failed to detect “not MBC” in 6.57% of all tests (6/70). However, the FCM results were within ±1 doubling dilution of the standard method, agreement between methods being 98.5% of all measurements. Thus, the label-free FCM can rapidly and accurately provide disinfectant-efficacy-testing results, with sensitivity and specificity compared with current standard efficacy tests.

FCM-based methods combined with fluorescent dyes were developed for the evaluation of disinfectant bactericidal efficacy [64,67,68]. However, the studies indicated that dye–bacterial components–antimicrobial agent interactions may affect the interpretation of the results. For example, it was shown that PI fluorescence was altered by chlorine as HOCl ions damage nucleic acids [67,69]. This approach allows the much earlier detection of a response compared with standard suspension tests for disinfectant-efficacy evaluation. The label-free FCM protocol is a very rapid and accurate method for the assessment of efficacy of disinfectants, requiring 4 h with excellent correlation with the reference tests.

The current standard culture-based tests, referenced in EN standard 13727+A2 [39], have major drawbacks, including being labor-intensive, time-consuming (take as long as 48 h to finalize) and providing only limited information on the impact of chemical disinfectants on microbial cells at low concentrations. A key feature of FCM is the ability to provide insights into the response of individual bacterial cells to chemical agents. By using different fluorescent dyes, the FCM is able to determine disinfectant-induced VBNC bacteria [9,67,70]. *Legionella* spp., several bacteria, including *E. coli*, *Enterococcus* spp., *P. aeruginosa*, *Helicobacter pylori* [41] were demonstrated to undergo VBNC states during UV or chlorine disinfection [52,71] that may underestimate the health risks associated with potential resuscitation of these forms. In the current study, the FCM method was demonstrated to provide insights into such injured bacterial populations at the disinfectant’s sub-inhibitory concentrations [9]. FCM, in combination with LIVE/DEAD^TM^ dual staining, identified *E. coli* in the VBNC state following treatment with Clor2Klin^TM^ at sub-inhibitory concentrations. Chlorine, the active ingredient of the commercial disinfectant, was reported to induce a VBNC state in *E. coli* during conventional disinfection processes [41,44,70,71,72]. Therefore, a potential application of this FCM-based method would be to assess the number of VBNC states during efficacy testing of various chemical disinfectants. The ability to detect VBNC bacteria has important implications for infection-control policies in hospital environments. Traditional culture-based methods may fail to identify VBNC pathogens, potentially underestimating microbial presence on surfaces or medical equipment. By using techniques such as label-free FCM to detect these viable but non-culturable cells, hospitals can improve the accuracy of microbial risk assessments. This could lead to more effective disinfection protocols, better-informed hygiene practices, and ultimately a reduction in HAIs.

The estimated cost of consumables and reagents for disinfectant-efficacy assessment using FCM is relatively low, approximately EUR 2.50 per sample (per microorganism) screened, excluding the costs associated with flow cytometry equipment and personnel [73]. However, as flow cytometers are commonly available in hematology and immunology laboratories, shared use by microbiology departments is a feasible and cost-efficient option.

There are certain known limitations to our investigation. Sample preparation and operation of FCM can be performed in a clinical microbiology lab without the need of additional training. However, interpretation of FCM data will require the automation of data analysis. Another important limitation of our study is the limited range of organisms tested, underscoring the need for future research to expand the scope. The performance of the label-free FCM-based method was only assessed against bacteria belonging to ESKAPEE group. Other microorganisms that have been described as persisting in hospital environments will need future studies. In addition, while scattered light scatter signals can provide insight into cell size and granularity, they are limited in their ability to accurately assess cell viability. These signals are often influenced by a range of factors unrelated to cell death, including sample preparation, cell activation, and instrument settings. In contrast, fluorescence-based assays provide a more specific method for evaluating cell viability, as they target biochemical markers associated with membrane integrity and apoptotic processes.

In conclusion, comparing standard suspension tests with the FCM method clearly demonstrates that the FCM technique is faster, less labor-intensive, and offers more detailed insights into disinfectant efficacy. Therefore, the developed FCM method is highly recommended for assessing the antibacterial activity of novel disinfectant formulations. This proof-of-principle study is particularly relevant as it addresses the growing demand to complement traditional microbiological testing with innovative, high-throughput consensus methods that can later be adopted by industry. The FCM analysis offers significant potential to enhance and expedite the assessment of chemical disinfectant efficacy.

## 5. Conclusions

Disinfectants play a central role in infection prevention and control, particularly in healthcare settings where MDR bacteria pose a major challenge. Among the most widely used classes of disinfectants globally are alcohols (such as ethanol and isopropanol), chlorine-based compounds (e.g., sodium hypochlorite), quaternary ammonium compounds (QACs), aldehydes (notably glutaraldehyde), and oxidizing agents like hydrogen peroxide and peracetic acid [74,75]. These agents have demonstrated broad-spectrum antimicrobial activity, including efficacy against Gram-positive and Gram-negative bacteria, enveloped viruses, and in some cases, bacterial spores. Recent global surveys and European healthcare guidelines highlight these compounds as the backbone of disinfection protocols [75,76], forming the scientific basis for selecting representative disinfectants for resistance assessment. The disinfectants Mikrozid™, Klinosept™, Glutanol™, Peroklin™, and Clor2Klin™ reflect this spectrum of chemical classes: Mikrozid™ and Klinosept™ are alcohol- or QAC-based formulations, Glutanol™ contains glutaraldehyde, Peroklin™ relies on peracetic acid and hydrogen peroxide, and Clor2Klin™ is a chlorine-based disinfectant. Evaluating their activity against resistant bacterial strains is essential, given concerns that improper or overuse of disinfectants could contribute to the selection of biocide-tolerant or cross-resistant microorganisms [77]. This study thus builds on established evidence regarding the global relevance and mechanisms of these key disinfectant classes to assess their comparative performance in combating resistant pathogens and provides new data on the predictive power of the label-free FCM method in assessing the efficacy of chemical disinfectants. Using five commercial disinfectants with different modes of action, the label-free FCM protocol accurately assessed the bactericidal efficacy of various chemical disinfectants, within four hours. Unlike traditional culture-based methods, which are time-consuming and labor-intensive, FCM rapidly, on the same-day, detects bacterial responses by analyzing scatter light profiles and count rates. The method demonstrated a strong agreement (91.42%) with standard tests, with high sensitivity (0.94) and specificity (0.98). Additionally, FCM identified bacteria in a viable but non-culturable (VBNC) state, providing critical insights into bacterial survival under sub-inhibitory disinfectant concentrations. Given its speed, accuracy, and ability to detect bacterial injuries at a single-cell level, the FCM method is a highly valuable tool for evaluating the bactericidal efficacy of new disinfectant formulations.

## Figures and Tables

**Figure 1 microorganisms-13-01156-f001:**
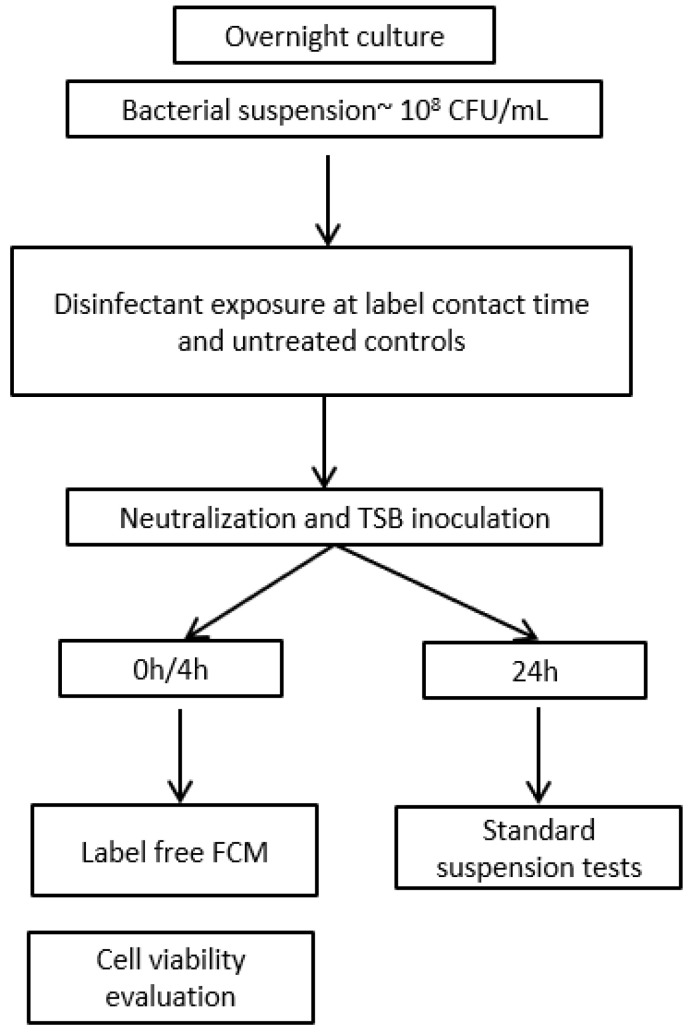
Experimental workflow for assessing the efficacy of the chemical disinfectants using both label-free FCM and standard suspension tests. Overnight bacterial cultures were used to prepare bacterial suspensions that were then put in contact with a range of two-fold dilutions of the chemical disinfectant, at label contact time. Bacterial suspensions without treatment (controls) were included. Disinfectant-treated samples and controls were evaluated using both the standard suspension tests and label-free FCM. Cell viability was determined using FCM with fluorescent labelling.

**Figure 2 microorganisms-13-01156-f002:**
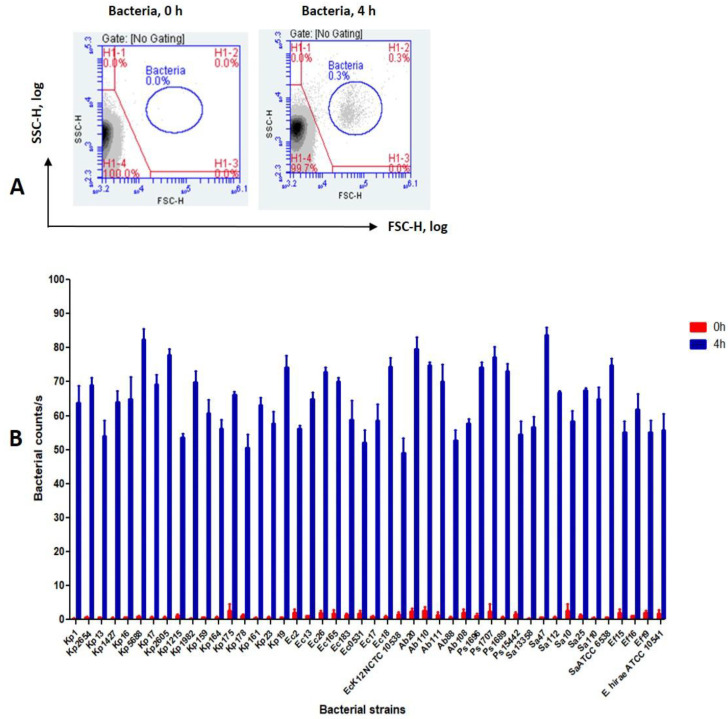
Changes in count rate of untreated bacterial strains after 4h incubation time. (**A**) Blue polygon: bacterial population boundary at 0 h and 4 h. (**B**) Flow cytometric bacteria events per second with cytometer set on medium, at 0 h (red bars) and 4 h (blue bars).

**Figure 3 microorganisms-13-01156-f003:**
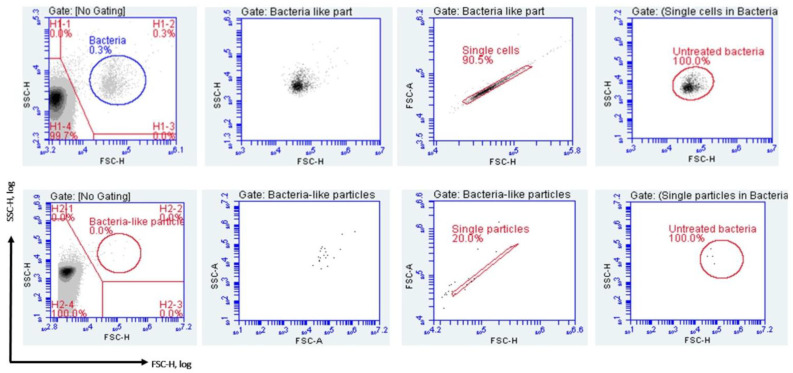
Label-free FCM-gating strategy for identification of bacteria-like particles. Upper panel: For each assay, the FSC and SSC signals of the untreated bacteria (growth control) were used to distinguish bacterial cells from instrumental noise and to set the gate on an FSC/SSC log-scale. The sample was back-gated to confirm that non-bacterial events were excluded. A bivariate plot (FSC-A vs. FSC-H) was used to exclude aggregates and to further select only single particles. A gate was then set for bacteria-like events, which was referred to as the untreated phenotype, and this gate was applied to all disinfectant-treated samples in the assay. Lower panel: The negative control (NC) plot shows that no events were detected within the bacteria-like particles gate.

**Figure 4 microorganisms-13-01156-f004:**
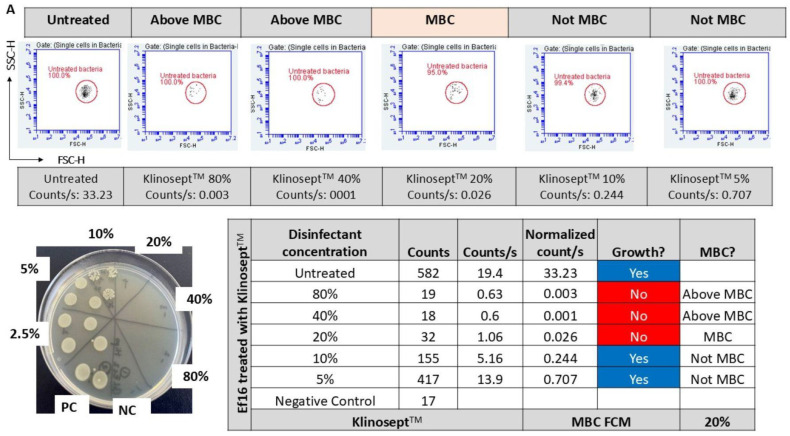
Disinfectant susceptibility-associated signatures and quantitative analysis of the FCM-derived counts for Gram-negative (**A**) and Gram-positive (**B**) strains are presented as representative results.

**Figure 5 microorganisms-13-01156-f005:**
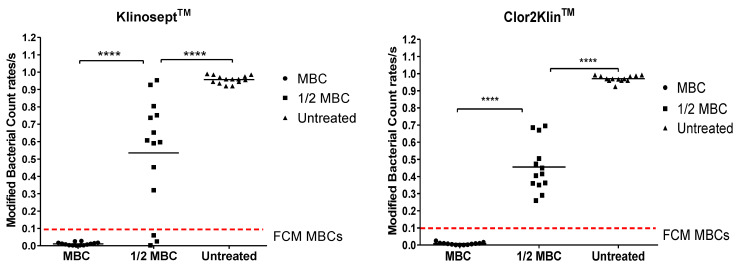
FCM analysis of bacterial cell counts/s for disinfectant-treated samples at minimum bactericidal concentration (MBC), subinhibitory concentration (½ MBC), and untreated growth controls across 14 strains. For each disinfectant, FCM counts differed significantly among treatment groups (Kruskal–Wallis test, ^****^*p* < 0.0001). Red dotted line indicates the empirically determined cut-off of <0.1 counts/s, as established by ROC curve analysis, used to classify FCM counts/s as “above MBC” or “not MBC”.

**Figure 6 microorganisms-13-01156-f006:**
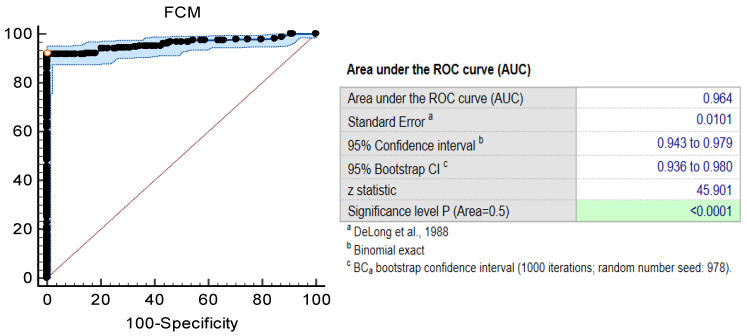
FCM measurements (counts/s) predict bactericidal efficiency of alcohol-, hydrogen-peroxide-, and chlorine-based disinfectants. The FCM method showed excellent prediction of disinfectant bactericidal efficiency when a threshold of ≤0.1 counts/s was applied (sens 0.92, spec 0.98).

**Figure 7 microorganisms-13-01156-f007:**
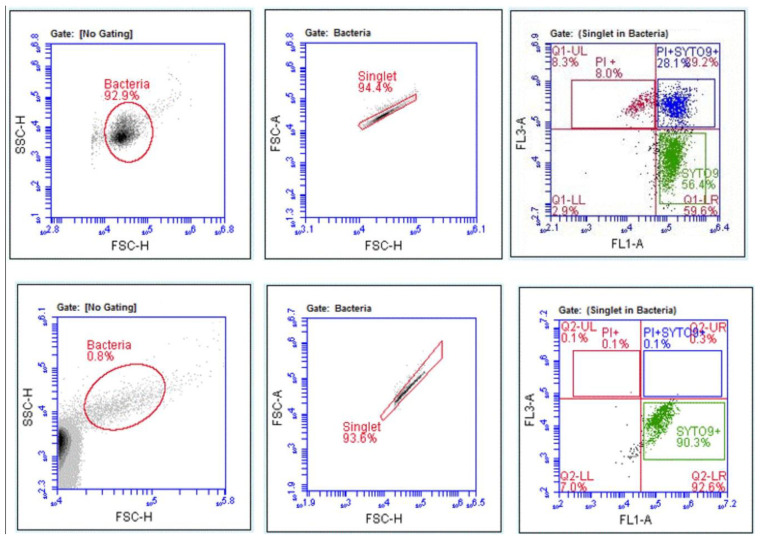
Upper panel: Viable bacteria were identified in disinfectant-treated samples at sub-inhibitory concentrations. Lower panel: Flow cytometry analysis of *E. coli* (Ec2) sample treated with disinfectant sub-inhibitory concentration and stained with LIVE/DEAD ^TM^ BacLight bacterial viability kit. The PI+ population corresponds to the cells stained with propidium iodine red fluorescent signal (FL2 at 630 nm; dead cells). The SYTO9+ population corresponds to green fluorescent signal (FL1 at 520 nm; live cells). The PI+SYTO9+ population (blue) corresponds to the double-stained cells (membrane-compromised cells or VBNC).

**Table 1 microorganisms-13-01156-t001:** Increased tolerance of microorganisms to chemical disinfectants.

Bacterial Species	Chemical Disinfectant	Mechanisms of Acquired Disinfectant Resistance	References
*Acinetobacter baumannii*	Triclosan	Efflux pumps FabI, AdelIJK	[23]
*Bacillus cereus*	Alcohol	Biofilms	[24]
*Bacillus* and *Staphylococcus*	Chlorine	ND	[25]
*Campylobacter* spp.	Benzalkonium chloride Chlorhexidine; Cetylpyridinium chloride	Efflux pumps	[26]
*Escherichia coli*	Hydrogen peroxide	RNA polymerase (rpo)	[27]
*E. coli*	Benzalkonium chloride	Efflux pumps qacE∆1gene	[28,29]
*E. coli*	Benzalkonium chloride	Membrane alterations	[30]
*Enteroccocus faecium*	Alcohol	ND	[31]
*E. coli*	Chlorophene Povidone-iodine	Porins OmpR; EnvZ	[27]
*Mycobacterium smegmatis*	Triclosan	Lipid metabolism (InhA)	[32]
*Mycobacterium chelonae*	Glutaraldehyde	Porins Msp	[33]
*P. aeruginosa*	Sodium hypochlorite	Efflux pumps MuxABC–OpmBa	[34]
*P. aeruginosa*	Chlorine	Efflux pumps, class 1 integrons intI1 gene	[35]
*P. aeruginosa*	Benzalkonium chloride	Efflux pumps *sugE*-A	[36]
*Staphylococcus epidermidis*	Benzalkonium chloride	Efflux pumps qacC/smr	[37]
*Salmonella enterica serovar enteritidis*	Benzalkonium chloride	Membrane alterations	[38]

**Table 2 microorganisms-13-01156-t002:** Chemical disinfectants reported to induce VBNC state in microorganisms.

Bacterial Species	Chemical Disinfectants	References
*E. coli,* *P. aeruginosa* *E. faecalis* *B. sphaericus* *Achromobacter* *Listeria monocytogenes* *Salmonella enterica* *Helicobacter pylori*	Chlorine	[41,42,43,44,45,46]
*L. monocytogenes*	Peracetic acid Hydrogen peroxide Quaternary ammonium	[47] [48]
*L. monocytogenes* *E. coli*	Peroxyacetic acid Chlorine dioxide	[49]
*Salmonella enterica*	Quaternary ammonium salt 75% ethanol Peracetic acid	[50]
*Yersinia enterocolitica*	Chloride	[51]
*Legionella* spp.	Monochloramine	[52]

**Table 3 microorganisms-13-01156-t003:** Active substances and label contact time and label concentrations for the chemical disinfectants tested.

Chemical Disinfectants Tested	Active Substances	Label Concentration	Label Contact Time
Mikrozid^TM^	25% ethanol, 35% Propan-1-ol	RTU	1 min
Klinosept^TM^	85% ethanol	RTU	1 min
Glutanol^TM^	2.4% glutaraldehyde, 15% ethanol	RTU	5 min
Peroklin^TM^	6.0% hydrogen peroxide	RTU	5 min
Clor2Klin^TM^	1.5% chlorine	1–10%	15 min

RTU = ready to use, label concentration = the concentration recommended by the manufacturer; label-contact time = the exposure time recommended by the manufacturer. Mikrozid^TM^—lot 1586747; Peroklin^TM^—lot 01220022R01; Clor2Klin^TM^—5949088105510; Glutanol^TM^—lot 01100805R01; Klinospet^TM^—4818BIO/02/12/24.

**Table 4 microorganisms-13-01156-t004:** Characteristics of bacteria used for disinfectant efficacy testing. The ESKAPEE bacterial strains were isolated from different clinical specimens (sputum, urine, skin infections, tracheal secretion, blood culture). The antibiotic resistance patterns of the clinical isolates as determined by the standard disk diffusion technique following the Clinical and Laboratory Standards Institute guidelines [53] against different antibiotics are presented. The majority of the clinical strains (9/10) were MDR.

Species	Strain Code	Source	Antibiotic Resistance Patterns
*K. pneumoniae*	Kp16	Sputum	CZ, AMP, E, MEM, IMP, CRO, CXM, FEP, AMC, FOX, CN, TE, CIP, TOB, SXT
*E. coli*	Ec2	Urine	AMC
*E. coli*	Ec17	Urine	CZ, PRL, AMP, CTX, ATM, CXM, FEP, AMC, TE, CIP, SXT
*E. coli*	Ec10538	ATCC reference strain	-
*A. baumannii*	Ab88	Sputum	SAM, IMP, MEM, DOR, FEP, ATM, CN, AK, CIP, CAZ
*A. baumannii*	Ab108	Skin infection	IMP, MEM, DOR, FEP, CN, AK, CIP, CAZ
*P. aeruginosa*	Ps1696	Secretion	TZP, MEM, IMP, FEP, ATM, CAZ, AK, DOR, CN, CIP, TOB
*P. aeruginosa*	Ps1707	Tracheal secretion	CAZ, ATM, FEP, MEM, IMP, AK, TOB, CIP, CN, DOR
*P. aeruginosa*	Ps15442	ATCC reference strain	-
*S. aureus*	Sa13358	Blood culture	FOX, AZM, P, E, TE
*S. aureus*	Sa47	Skin infection	FOX, AZM, P, CN, E, TE, LZD
*S. aureus*	Sa6538	ATCC reference strain	-
*E. faecium*	Ef16	Tracheal secretion	TE, CN, CIP, P, VA
*E. hirae*	Eh10541	ATCC reference strain	-

Legend: SAM ampicillin/sulbactam, CZ cephazolin, AMC amoxicillin-clavulanic acid, FOX cefoxitin, AZM azithromycin, P penicillin, TE tetracycline, CN gentamycin, E erythromycin, CIP ciprofloxacin, ATM aztreonam, DOR doripenem, AK amikacin, FEP cefepime, PRL piperacillin, MEM meropenem, IMP imipenem, LZD linezolid, CXM cefuroxime, CAZ ceftazidime, CRO ceftriaxone, TOB tobramycin, TZP piperacillin-tazobactam, VA vancomycin, SXT trimethoprim sulfamethoxazole, AMP ampicillin.

**Table 5 microorganisms-13-01156-t005:** The median bactericidal concentrations (%) resulting in a logarithmic reduction ≥ 5 as determined by the quantitative suspension tests. Logarithmic reductions (LR) in viable counts were calculated in accordance with SR EN 13727+A2. The bactericidal concentrations (%) of each disinfectant that achieved a logarithmic reduction (LR) ≥ 5 in viable counts against the tested bacterial strains are reported.

Bacterial Strains	Chemical Disinfectant				
	Mikrozid^TM^ (%)	Klinosept^TM^ (%)	Glutanol^TM^ (%)	Peroklin^TM^ (%)	Clor2Klin^TM^ (%)
Kp16	30	30	1.25	0.07	0.004
Ec2	30	60	2.5	0.15	0.003
Ec17	30	60	2.5	0.15	0.003
Ec10538	30	40	1.25	0.15	0.004
Ab88	15	30	2.5	0.07	0.001
Ab108	15	30	2.5	0.07	0.001
Ps1696	10	20	1.25	0.47	0.003
Ps1707	10	20	1.25	1.87	0.003
Ps15442	30	40	2.5	0.07	0.003
Sa13358	40	40	0.62	0.07	0.006
Sa47	40	60	1.25	0.07	0.003
Sa6538	60	40	2.5	0.07	0.025
Ef16	30	30	5	0.07	0.001
Eh10541	30	40	2.5	0.07	0.003

Kp, *Klebsiella pneumoniae*; Ec, *Escherichia coli*; Ab, *Acinetobacter baumannii*; Sa, *Staphylococcus aureus*; Ef, *Enterococcus faecium*; Eh, *Enterococcus hirae*; Ps, *Pseudomonas aeruginosa*.

**Table 6 microorganisms-13-01156-t006:** Agreement between the standard suspension tests and label-free optical FCM method applied for alcohols, oxidizing and alkylating agents. A total of 10 clinical and 4 reference strains of *K. pneumoniae* (*n* = 1), *E. coli* (*n* = 3), *P. aeruginosa* (*n* = 3), *A. baumannii* (*n* = 2), *S. aureus* (*n* = 3), *Enterococcus* spp. (*n* = 2) were each tested against the various disinfectants, with different mechanisms of action: alcohol-, hydrogen-peroxide-, glutaraldehyde- and chlorine-based disinfectants.

Bacterial Strains		Chemical Disinfectants				
		Mikrozid^TM^	Klinosept^TM^	Glutanol^TM^	Peroklin^TM^	Clor2Klin^TM^
Kp16						
Ec2						
Ec17						
EcK12 NCTC 10538						
Ab88						
Ab108						
Ps1696						
Ps1707						
Ps15442						
Sa13358						
Sa47						
SaATCC 6538						
Ef16						
*Eh* ATCC 10541						
Agreement		12/14 (85.71%)	11/14 (78.57%)	14/14 100 (%)	13/14 (92.85%)	14/14 (100%)
Agreement within ±1 dilution		14/14 (100%)	13/14 (92.85%)	14/14 (100%)	14/14 (100%)	14/14 (100%)
Agreement within ±2 dilution		14/14 (100%)	14/14 (100%)	14/14 (100%)	14/14 (100%)	14/14 (100%)
	Agreement					
	Error is within ±1 dilution					
	Error is within ±2 dilution					

Kp, *Klebsiella pneumoniae*; Ec, *Escherichia coli*; Ab, *Acinetobacter baumannii*; Sa, *Staphylococcus aureus*; Ef, *Enterococcus faecium*; Eh, *Enterococcus hirae*; Ps, *Pseudomonas aeruginosa*.

## Data Availability

The FCS files corresponding to the Figure 3, Figure 4 and Figure 7 are public available at https://community.cytobank.org/cytobank/experiments/115822 (accessed on 24 April 2025) (FCM for bactericidal efficacy of disinfectants). All the FCS files are available upon request from the corresponding author, Luminita Marutescu via email (luminita.marutescu@bio.unibuc.ro).

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
