# Peer review of "Label-Free Flow Cytometry: A Powerful Tool to Rapidly and Accurately Assess the Efficacy of Chemical Disinfectants"

_microorganisms, 2025, doi:10.3390/microorganisms13051156_

Round 1
Reviewer 1 Report
Comments and Suggestions for Authors
The manuscript has several errors, making it unacceptable for publication in its current state.
- In Abstract, "The label-free FCM provided the results in approximatively 4 hours and showed strong correlation with standard tests (over 9.42% agreement), that can take up to 48 hours." 9.42% agreement is not a strong correlation; I couldn't determine how the number was obtained. Unless it came from lines 302-304, "When this threshold was applied to the tested samples, 64 out of 70 302 FCM results (91.4% of FCM assay results) were classified correctly for "above or MBC" 303 status (sensitivity 0.94 and specificity 0.98) (Table 4, Figure 6)." If this is the case, it should be 91.4% agreement.
- Line 84, "A total of 16 bacterial strains..."; however, in the following text and Table 2, only 14 were listed.
- Line 89, "The clinical strains were recovered from different sources 88 of infection (Table 2) and the majority (36/42) were multi drug resistant (MDR)." If 10 clinical bacteria were used in this study, it was unclear where "36/42" came from. Maybe I missed some information?
- Table 3 listed 13 strains, and one strain was missing.
- Fig 2 B showed more bacterial strains, which were not mentioned in Table 2.
- Fig 5 shows the differences in FCM counts/s assayed for MBC, ½ MBC, and untreated bacteria. However, whether the data were the average of all 14 strains or the representative data of one bacterial strain is unclear. I assume it should be the latter, but I couldn't find more info.
Additionally, only two representative data were shown in Fig 4: Ef16/Klinosept and Ec2/Peroklin. The authors should provide all 70 data sets (i.e., 14 strains x 5 disinfectants) in Supplementary Materials. They did give a link to the FCS files; however, the data were minimal—only related to what was presented in Figs 3, 4, and 7.
Author Response
We are deeply grateful to the Reviewer 1 for thorough and thoughtful assessments of our manuscript. Their expert insights and valuable recommendations have significantly contributed to refining our study, and we appreciate the time and effort they dedicated to this process.
Please refer to the attached document for detailed responses to the reviewer’ comments, with the corresponding revisions and corrections highlighted in the resubmitted files.

Reviewer 2 Report
Comments and Suggestions for Authors
Thank you for the opportunity to review your research investigating the potential of flow cytometry to assess efficacy of disinfectants. Please see my feedback below:
Abstract: Ln 24: 'over 9.42% agreement', I believe this is a typo?
Introduction: Ln 63-65. Citation 23 is not the correct citation for the Klebsiella information. Please find the appropriate citation.
Methods: Ln 108 - which equates to how many CFU? Please ensure the CFU is apparent for each step, or at least the first and final. Ln 126.
Figure 1. The steps following Neutralization and TSB inoculation should not differ between test group and controls, yet your flow diagram seems to suggest they do.
Results: 3.1 - why were the MIC values not provided?
Fig. 4. Please expand either the figure legend or the table to make it clear what each column denotes.
The data covered in figures 4 & 5 need explaining in more detail in the text.
Overall an interesting paper, addressing a key area of public health with suggestions to improve what we are currently doing. A brief discussion of relative cost in the discussion section would have been informative.
Author Response
We are deeply grateful to the Reviewer 2 for thorough and thoughtful assessments of our manuscript. Their expert insights and valuable recommendations have significantly contributed to refining our study, and we appreciate the time and effort they dedicated to this process.
Please refer to the attached document for detailed responses to the reviewer’ comments, with the corresponding revisions and corrections highlighted in the resubmitted files.
Thank you very much!
Kind regards!

Reviewer 3 Report
Comments and Suggestions for Authors
Viable but nonculturable bacteria (VBNC) have been shown to retain virulence even after certain disinfection procedures, constituting a considerable public health concern due to their non-detectability through conventional microbiological techniques. Flow cytometry (FCM) has emerged as a promising tool in the field of food microbiology. This technique facilitates the rapid identification of the various physiological states of bacteria following disinfection procedures, offering significant advantages in the analysis of microbial communities. In this study, the researchers employed a previously developed method, namely, label-free FCM, to expeditiously assess the bactericidal efficacy of various chemical disinfectants. These disinfectants possess differing mechanisms of action and are frequently utilized in clinical settings. The disinfectants in question include alcohols, oxidizing agents, and alkylating agents. Furthermore, the study investigated the potential of FCM in conjunction with fluorescent dyes for the detection of disinfectant-induced VBNC state, a phenomenon that eludes detection by conventional efficacy testing methods.
The research methods are chosen correctly and are described in sufficient detail. The results are discussed in detail. The patterns found in the study are hypothetically explained by the authors.
Major cocern.
A discussion of the method's limitations is warranted. Due to the large possible signal shifts, fluorescence-based viability tests are typically preferred over scattered light signals in flow cytometry.
Minor concern.
Fig.4 A: As indicated by the image presented, the 10% Klinosept solution is displayed on two occasions.
Author Response
We are deeply grateful to the Reviewer 3 for thorough and thoughtful assessments of our manuscript. Their expert insights and valuable recommendations have significantly contributed to refining our study, and we appreciate the time and effort they dedicated to this process.
Please refer to the attached document for detailed responses to the reviewer’ comments, with the corresponding revisions and corrections highlighted in the resubmitted files.
Thank you very much!
Kind regards!

Reviewer 4 Report
Comments and Suggestions for Authors
Title: Antimicrobial Resistance: Challenges and Innovative Solutions.
In this paper, the authors study the bactericidal efficacy of chemical disinfectants to ensure effective infection control. The bactericidal efficacy is measured by flow cytometry (FCM). The authors test five commercial disinfectants (alcohols, oxidizing agents and alkylating agents). The study concludes that the FCM method is a powerful tool to evaluate the efficacy of new disinfectant formulations.
-To facilitate the reader's understanding, the 3 tables that only present the title, and are very complicated, require a long and clear legend.
-Figure 1 and 4, could be better explained.
-This is a good piece of study however I have some concerns.
For example, some agents can be irritants and cause inflammation. The article, although good, lacks this important part that the authors should address even briefly. Therefore, to make this paper more interesting for the readers of this important journal, the authors should expand a bit the discussion (or introduction) on mechanism of immune response and inflammation. Below I report an interesting article that should be studied, incorporate the meaning and report it briefly in the list of references.
Saggini R, Pellegrino R. MAPK IS IMPLICATED IN SEPSIS, IMMUNITY, AND INFLAMMATION. International Journal of Infection. 2024;8(3):100-104. (www.biolife-publisher.it)
-In addition, authors should mention briefly the role of inflammatory and anti-inflammatory cytokines, when discussing inflammation. Again, below I report an interesting article that should be studied, incorporate their meaning and report it briefly in the discussion and in the list of references.
Toniato E. IL-37 is an inhibitory cytokine that could be useful for treating infections . International Journal of Infection. 2024;8(1):1-2. (www.biolife-publisher.it).
-I believe these suggestions are important for improving this paper. Without these corrections the paper cannot be published. So, I recommend minor revision.
could be better
Author Response
We are deeply grateful to the Reviewer 4 for thorough and thoughtful assessments of our manuscript. Their expert insights and valuable recommendations have significantly contributed to refining our study, and we appreciate the time and effort they dedicated to this process.
Please refer to the attached document for detailed responses to the reviewer’ comments, with the corresponding revisions and corrections highlighted in the resubmitted files.

Reviewer 5 Report
Comments and Suggestions for Authors
Introduction
You wrote: “Chemical disinfectants are widely used to reduce bacterial bioburden on various surfaces in medical environments. However, clinically important pathogens have been found to persist even after disinfection, posing a risk of infection [1-3]. Several studies, both experimental and real-world, have demonstrated bacteria's ability to adapt to chemical disinfectants [4, 5]. Outbreak investigations reveal that nosocomial pathogens can withstand exposure to disinfectants such as peracetic acid [6], quaternary ammonium compounds [7], and glutaraldehyde [8]. For instance, Serratia marcescens was not eliminated by a quaternary ammonium compound-based disinfectant, and Mycobacterium massiliense strains responsible for outbreaks in 38 hospitals in Rio de Janeiro, Brazil, exhibited resistance to glutaraldehyde, which was used for endoscope disinfection. These findings underscore the need for careful selection of disinfectants, routine environmental screening, and, when necessary, testing disinfectant tolerance in specific contexts [2, 6, 9 -11”
[This is the key reason for your study. To make your study more widely applicable please review the literature and report in detail all the studies which demonstrated resistance of specific pathogens to specific disinfectants, perhaps presenting these data also in a table]
You wrote: “Most importantly, studies have shown that chemical disinfectants can induce a viable but nonculturable (VBNC) state, in which bacteria remain alive but fail to grow on routine culture media. In contrast to these conventional methods, single-cell technologies such as flow cytometry (FCM) offer rapid analysis within minutes, require minimal sample volumes, and provide faster and more accurate detection of bacterial viability.”
[This is also key to the purpose of your study. Please review the literature and report all studies on non culturable but viable pathogens after disinfection]
Methods
Please explain why you chose each of these disinfectants for testing.
You wrote” “A total of 16 bacterial strains were used in this study, comprising 4 reference strains Staphylococcus aureus ATCC 6538, Pseudomonas aeruginosa ATCC 15442, Enterococcus hirae ATCC 10541 and Escherichia coli K12 NCTC 10538 and 10 clinical bacteria belonging to the ESKAPEE group [K. pneumoniae (n=1), E. coli (n=2), Acinetobacter (n=2), P. aeruginosa (n=2), S. aureus (n=2); E. faecium (n=1)]. The clinical strains were recovered from different sources of infection (Table 2) and the majority (36/42) were multi drug resistant (MDR).”
[Please explain the origin of these samples. How many are from patients in your hospital and are they representative of Romania?
[All of the figures contain data panels and labelling not legible at this magnification. You will need to spread your tables and figures over more than the pages you are currently using].
[Conclusion.
I agree with your conclusions. It would be good to know if your results reflect the overall usage of these specific chemicals in e.g. Europe or the ECU or the US…]
The request for major revision is for the improvement of the legibility of the graphs, tables and labelling.
Author Response
We are deeply grateful to the Reviewer 5 for thorough and thoughtful assessments of our manuscript. Their expert insights and valuable recommendations have significantly contributed to refining our study, and we appreciate the time and effort they dedicated to this process.
Please refer to the attached document for detailed responses to the reviewer’ comments, with the corresponding revisions and corrections highlighted in the resubmitted files.

Reviewer 6 Report
Comments and Suggestions for Authors
My comments as follows.
The abstract is clear but could be made more impactful by briefly mentioning the main results (e.g., numerical accuracy, sensitivity, and specificity values) to emphasize the strength of the label-free FCM approach.
The introduction is well written; however, the authors could expand slightly on the limitations of traditional culture-based methods versus FCM at the beginning to immediately highlight the novelty of the study.
In Section 2.1, please include more detailed information about the lot numbers or batch numbers of the disinfectants used, if possible, to ensure reproducibility.
The manuscript discusses clinical and reference strains. It would be beneficial to add the year of isolation or clinical relevance (hospital-acquired, community-acquired) of the clinical strains for better context.
In Section 2.4.3, the flow cytometer settings are described, but please clarify whether compensation controls for fluorescent signals were applied or if spectral overlap was negligible.
The statistical methods are appropriate; however, the authors should mention if normality was assessed before applying the unpaired t-test with Welch’s correction.
Table 3 provides important data, but it would strengthen the paper if the authors briefly discussed any trends observed (e.g., why Clor2Klin showed higher efficacy across strains) directly in the results section.
Some figures (e.g., Figure 3 and Figure 7) could benefit from clearer labeling of axes and gating regions. Consider increasing font size for better readability.
The comparison between FCM and standard methods is strong. However, it would add value if the authors included an additional small table summarizing false positives and false negatives identified by the FCM method.
The discussion of VBNC detection is interesting. The authors could improve this section by briefly mentioning potential implications for infection control policies.
The manuscript would benefit from a short paragraph discussing limitations, such as the potential variability introduced by different flow cytometer models or sample preparation steps.
References are generally relevant and recent. However, a few references on newer FCM applications for bacterial viability assessment (published in 2023–2024) could strengthen the background discussion.
Overall, the language is good, but there are minor grammatical errors and inconsistencies (e.g., inconsistent spacing, occasional typos like "in light of disinfectant’ efficacy" instead of "disinfectant’s efficacy"). A careful proofreading is recommended.
The availability of raw FCM data is excellent. It would be helpful if the authors could briefly mention in the main text how readers could access and utilize the supplementary files for independent verification.
Overall Recommendation:
The work has significant merit and presents well structures. However, the manuscript requires minor revisions, before it can be accepted.
Comments on the Quality of English LanguageOverall, the language is good, but there are minor grammatical errors and inconsistencies. A careful proofreading is recommended.
Author Response
We are deeply grateful to the Reviewer 6 for thorough and thoughtful assessments of our manuscript. Their expert insights and valuable recommendations have significantly contributed to refining our study, and we appreciate the time and effort they dedicated to this process.
Please refer to the attached document for detailed responses to the reviewer’ comments, with the corresponding revisions and corrections highlighted in the resubmitted files.

Round 2
Reviewer 1 Report
Comments and Suggestions for Authors
Lines 356-360 are redundant.
Author Response
Thank you for your comment. We agree that lines 356–360 contain redundant information. We have revised the section to remove repetition.
Reviewer 5 Report
Comments and Suggestions for Authors
This is now an outstanding article. Well done. Hooray! Thank you for all the corrections and additions. Tables are great and you noted the efflux pumps involved!! This is now a tribute to Romanian laboratory science.
Don't hesitate to spread your figures over even more pages (this is an electronic publication). If you bothered to compute the data they are worth being completely readable by future researchers who can then quote your results.
Great work! Romanian patients will now be safer.
Author Response
Thank you so much for your generous feedback!